# Factors Affecting Voriconazole Trough Concentration and Optimal Maintenance Voriconazole Dose in Chinese Children

**DOI:** 10.3390/antibiotics10121542

**Published:** 2021-12-16

**Authors:** Yi-Chang Zhao, Yang Zou, Jing-Jing Hou, Chen-Lin Xiao, Bi-Kui Zhang, Jia-Kai Li, Da-Xiong Xiang, Indy Sandaradura, Miao Yan

**Affiliations:** 1The Second Xiangya Hospital, Central South University, Changsha 410011, China; zhaoyichang@csu.edu.cn (Y.-C.Z.); houjingjingo@163.com (J.-J.H.); xiaochenlin@csu.edu.cn (C.-L.X.); 505995@csu.edu.cn (B.-K.Z.); lijiakai@csu.edu.cn (J.-K.L.); xiangdaxiong@csu.edu.cn (D.-X.X.); 2Institute of Pharmacy, Central South University, Changsha 410011, China; 3Institute of Clinical Pharmacy, Xiangtan Central Hospital, Xiangtan 410010, China; zy15298363690@stu.cpu.edu.cn; 4School of Medicine, University of Sydney, Sydney, NSW 2006, Australia; Indy.Sandaradura@health.nsw.gov.au; 5Centre for Infectious Diseases and Microbiology, Westmead Hospital, Sydney, NSW 2145, Australia

**Keywords:** voriconazole, children, maintenance dose, dosage regime, influencing factors, CYP2C19

## Abstract

Voriconazole is a triazole antifungal agent commonly used for the treatment and prevention of invasive aspergillosis (IA). However, the study of voriconazole's use in children is limited. The present study was performed to explore maintenance dose to optimize voriconazole dosage in children and the factors affecting voriconazole trough concentration. This is a non-interventional retrospective clinical study conducted from 1 January 2016 to 31 December 2020. The study finally included 94 children with 145 voriconazole trough concentrations. The probability of achieving a targeted concentration of 1.0–5.5 µg/mL with empiric dosing increased from 43 (45.3%) to 78 (53.8%) after the TDM-guided adjustment. To achieve targeted concentration, the overall target maintenance dose for the age group of less than 2, 2 to 6, 6 to 12, and 12 to 18 years old was approximately 5.71, 6.67, 5.08 and 3.31 $\mathrm{mg}\cdot {\mathrm{kg}}^{-1}/12\;h$, respectively (*p* < 0.001). Final multivariate analysis found that weight (*p* = 0.019), dose before sampling (*p* < 0.001), direct bilirubin (*p* < 0.001), urea nitrogen (*p* = 0.038) and phenotypes of CYP2C19 were influencing factors of voriconazole trough concentration. These factors can explain 36.2% of the variability in voriconazole trough concentration. Conclusion: In pediatric patients, voriconazole maintenance doses under the target concentration tend to be lower than the drug label recommended, but this still needs to be further studied. Age, body weight, dose, direct bilirubin, urea nitrogen and phenotypes of CYP2C19 were found to be influencing factors of voriconazole concentration in Chinese children. The influence of these factors should be taken into consideration during voriconazole use.

## 1. Introduction

Children with leukemia, prolonged neutropenia, receiving high-dose corticosteroids, and those undergoing allogeneic hematopoietic stem cell transplants are at a high risk of invasive fungal infection (IFI) with high morbidity and mortality [1,2]. Voriconazole could be used for the prophylaxis and treatment of invasive aspergillosis and candidiasis in non-neutropenia patients. Some serious invasive infections caused by fluconazole-resistant candida (including Candida glabrata)) and the genera of Scedosporium and Fusarium can also be treated by voriconazole [3], a broad-spectrum second-generation triazole antifungal agent [4,5] that is also recommended as the first-line treatment of invasive aspergillosis [6].

Voriconazole is both a substrate and a potent inhibitor of cytochrome P450 (CYP) 2C19 and 3A. It is mainly metabolized to voriconazole-N-oxide, and exhibits non-linear pharmacokinetics and its plasma concentration increases disproportionally with dosage escalating in most cases [7,8,9,10,11]; however, a multicenter study about the parenteral formulation of voriconazole in immunocompromised pediatric patients (2 to 11 years old) found that elimination of voriconazole was linear in children following doses of 3 and 4 mg/kg every 12 h [12]. In addition, a recent study about population pharmacokinetic analysis conducted on pooled data from 112 immunocompromised children (2 to <12 years), 26 immunocompromised adolescents (12 to <17 years), and 35 healthy adults found that a two-compartment model with first-order absorption and mixed linear and nonlinear (Michaelis–Menten) elimination adequately described the voriconazole data [13]. Therefore, the population pharmacokinetic characteristics of voriconazole are relatively complex.

Voriconazole has a narrow therapeutic index with numerous studies also showing high steady-state plasma concentrations to be associated with clinical adverse events, whilst inadequate concentration was more likely to result in treatment failure [4,14,15]. Several factors, including age, body weight, cytochrome P450 2C19 genotype, concomitant drugs, liver function, and food, are responsible for the large variability in voriconazole metabolism [8,13,16].

Clinical Pharmacogenetics Implementation Consortium (CPIC) Guidelines for CYP2C19 and voriconazole therapy reported that there was substantial evidence linking CYP2C19 genotype with phenotypic variability in voriconazole pharmacokinetics; however, the paucity of studies and their inconsistent findings prevented the authors from making a recommendation for patients with multiple gene polymorphisms [17]. The voriconazole guideline by Chinese Pharmacological Society had no recommendation on the benefits of the CYP2C19 gene test before initiating voriconazole regarding its efficacy and safety [18]. As a result, the influence of CYP2C19 genotype on voriconazole use in pediatric patients still needs to be further explored.

In order to maximize efficacy and reduce the risk of adverse reactions, routine therapeutic drug monitoring (TDM) of voriconazole trough concentrations is strongly recommended [17,18,19]. To achieve targeted voriconazole concentrations (1.0–5.5 mg/L), 7 to 8 mg/kg intravenous twice daily were recommended for Caucasian children [20,21], while in Chinese children, it was used mainly according to the drug label. Follow the instructions of the “VFEND” drug label approved by the U.S. Food and Drug Administration in 2021, children 2 to 14 years of age weighing less than 50 kg are recommended a loading dose of 9 mg/kg and a maintenance dose of 8 mg/kg for intravenous use, while for oral use, the maintenance dose is also 9 mg/kg (maximum dose of 350 mg every 12 h) [3]. For children aged 12 to 14 years weighing greater than or equal to 50 kg and those aged 15 years and older, regardless of body weight, adult dosages are recommended. The adult loading dose and maintenance dose for intravenous use is only 6 mg/kg and 4 mg/kg, respectively [3]. High variability and less literature on recommended doses in voriconazole exposure suggest that it is necessary to optimize the voriconazole dose, especially in pediatric patients.

To sum up, the purpose of this article is to evaluate voriconazole dosing and concentration in different age groups. Voriconazole maintenance doses under the target concentration tend to be lower than drug label recommended; however, this still needs to be further studied. Simultaneously, it aims to identify the major factors responsible for the inter-individual variability of voriconazole and critically analyze the impact of the influencing factors on pediatric patients.

## 2. Results

### 2.1. Demographics and Clinical Characteristics

Ninety-four children with a total of 145 voriconazole trough concentrations were included in the study. Demographics, clinical characteristics, CYP2C19 genotypes and drug combinations are presented in Table 1 according to the age groups of ≤2, 2–6, 6–12 and 12–18 years old. The primary diagnosis was hematological malignancy (76.6%), followed by respiratory infection (69.1%) and bloodstream infection (38.3%). The major infection sites were lung (66.0%) and blood (25.5%). Among the enrolled patients, most (*n* = 48; 51.1%) were administrated prophylactic voriconazole, together with therapeutically (*n* = 19; 20.2%) and empirically (*n* = 27; 28.7%), respectively. 15 (19.0%) patients were diagnosed with confirmed IFI according to European Organisation for Research and Treatment of Cancer (EORTC) criteria. Three (3.18%) patients experienced voriconazole-related adverse reactions, two of them with persistent elevation of alanine aminotransferase and/or aspartate aminotransferase and vomiting just in one. Of the two patients with liver function derangement, one patient with a trough concentration of 5.32 µg/mL gradually recovered after drug withdrawal, and the other recovered following a reduction in voriconazole dosage. For the patient with vomiting, symptoms improved after conversion from oral to intravenous administration.

### 2.2. Measurement of Plasma Voriconazole Concentrations and Dosing: Empiric and TDM-Guided

The voriconazole trough concentration results between the four age groups of less than 2, 2–6, 6–12, and 12–18 years old are shown in Table 2. The median initial trough concentration of the four groups were 0.17, 0.87, 2.45 and 2.15 µg/mL (*p* = 0.0014; Figure 1a), while the median overall concentrations were 0.18, 1.19, 2.02 and 2.02 µg/mL, respectively (*p* = 0.0096; Figure 1b). Further pairwise comparison analysis about the initial and overall concentration between each two groups is shown in Table 3. For all the study patients, initial probability of achieving targeted concentration of 1.0–5.5 µg/mL with empiric dosing was 43 (45.3%), while after the TDM-guided adjustment, the overall probability of achieving targeted concentration increased to 78 (53.8%). Initial and overall distributions of the four age groups are shown in Figure 1c,d. For the age group of less than 2 years old, 60.0% of the children were subtherapeutic at the initial dosing, only 40.0% of children achieved the target concentration. The proportion of the children who failed to achieve targets decreased as their age increased, while only 20.0% of the children were subtherapeutic at the group of 12–18 years old (*p* = 0.004; Figure 1c). Meanwhile, the overall subtherapeutic percentage of the four groups was 61.1%, 45.5%, 28.1% and 23.0% (*p* < 0.001; Figure 1d).

The initial mean maintenance doses for the four age groups were 7.10, 6.30, 5.20, and 3.35 mg/kg twice daily, respectively, while the TDM-guided median maintenance doses were 7.14, 6.67, 5.10 and 3.60 mg/kg twice daily (Figure 2a). Utilizing the target concentration range of 1.0–5.5 µg/mL, the initial target maintenance doses for the four age groups were 5.75, 6.90, 5.10 and 3.30 mg/kg twice daily, while the overall target maintenance dose were 5.71, 6.67, 5.09 and 3.31 mg/kg twice daily (Figure 2b), respectively. Further pairwise comparison analysis about the overall and targeted maintenance doses by Bonferroni adjustment is also shown in Table 3.

### 2.3. Variability of Voriconazole Concentrations

Voriconazole TDM was performed only once in 59 (62.8%) children, twice in 28 (29.8%) children and three or more times in 7 (7.45%) children. The variation of voriconazole was analyzed among 35 (37.2%) children who received two or more TDM measurements. The coefficient of variation ranged from 17.4% to 143.0%. The inter-individual coefficient of variation ranged from 1.68% to 678.5%. Scatter diagrams and inter-patient variability of overall voriconazole trough concentration at different weight-adjusted doses and overall maintenance dose within the target concentration range were shown in Figure 2c,d, respectively. In addition, in the scatter diagram of Figure 2d, the concentration is widely distributed in children less than 12 years old, while it is intensively distributed in children older than 12 years old. Therefore, children less than 12 years old seemed to have a higher variability.

### 2.4. Factors Affecting Voriconazole Trough Concentrations

Correlation analysis was performed to analyze the correlation between voriconazole trough concentration and various factors. Age, body weight, height, dose before sampling, aspartate aminotransferase, total bilirubin, direct bilirubin, albumin, urea nitrogen, creatinine, genotypes and phenotypes of CYP2C19 were found to be significant. The results are shown in Table 4.

Subsequently, multivariate analysis by linear regression was performed to analyze the effect of these factors on voriconazole trough concentrations. The coefficients of multivariate analysis by linear regression model are shown in Table 5. Weight, dose before sampling, direct bilirubin, urea nitrogen and phenotypes of CYP2C19 were remained in the final model. The linear relationship between trough concentrations and these independent variables was significant (F = 8.551, *p* < 0.001). The model could explain 36.2% of the variability in trough concentrations.

The multiple linear regression equation was as follows:Ctrough = −0.655 − 0.050 * weight + 0.033 * dose + 0.055 * direct bilirubin + 0.216 * urea nitrogen − 1.789 * CYP2C19 intermediate metabolizer * A − 1.521 * CYP2C19 normal metabolizer * B(1)

(“A = 1” if the patient is classified as CYP2C19-intermediate metabolizer, otherwise “A = 0”; “B = 1” if the patient is classified as CYP2C9-normal metabolizer; otherwise “B = 0”; Compared to CYP2C19 phenotype (PM) group, dealt with the operation of dummy variables).

### 2.5. Effect of Drug Combination

Concomitant use of tacrolimus (Figure 3) had no significant effect on voriconazole concentration, corrected concentration (concentration/dose) and unit kilogram maintenance dose, but it had a significant effect on maintenance dose.

Similar analysis was performed for the combination of glucocorticoids and proton pump inhibitors (PPIs). Concomitant use of glucocorticoids (Figure 4) had no significant effect on voriconazole concentration, corrected concentration, and maintenance dose; however, it had a significant effect on unit kilogram maintenance dose.

For the concomitant use of PPIs (Figure 5), it had no significant effect on voriconazole concentration, corrected concentration, and unit kilogram maintenance dose, but it had a significant effect on maintenance dose.

### 2.6. Effect of CYP2C19

Detailed analysis was performed among different genotypes and metabolic types. The *p* value of concentration, corrected concentration, maintenance dose and unit kilogram maintenance dose among 3 metabolic subgroups were 0.1565, 0.0031, 0.0694 and 0.0015 respectively (Figure 6). The poor metabolic type seemed to have the highest corrected concentration and lowest unit kilogram maintenance dose.

In the present study, we considered differences caused by different genotypes. The *p* value of concentration, corrected concentration, maintenance dose and unit kilogram maintenance dose among 5 genotype subgroups were 0.2738, 0.0337, 0.1737 and 0.0572, respectively (Figure 7); thus, different gene polymorphisms may also cause significant differences in correction concentrations.

## 3. Discussion

Data to support the safe use of voriconazole in children is sorely lacking, especially for children under 2 years old. This retrospective study provides insights into the use of voriconazole in children of different age groups and provides guidance for more precise dosing in Chinese children. Currently, for children younger than 12 years, voriconazole showed nonlinear pharmacokinetics [22,23], warranting that more cautious dose adjustment was necessary for this group. Our study found high variability in voriconazole trough levels, with mean intra- and inter-individual variation of 72.9% and 111.2%, respectively, similar to previous publications [9,24]. Furthermore, more than 50.0% of children did not reach the target range of 1.0–5.5 µg/mL with empiric dosing, also consistent with previous studies [25]. The initial trough concentration in children aged 2–12 was significantly higher than that in children aged under 2 years, suggesting faster metabolism in infants. In addition, the European Medicines Agency (EMA) approved higher doses in 2 to 12 years old children, an 8 mg/kg intravenously twice daily (9 mg/kg day^−1^) or a 9 mg/kg orally twice daily. Our data suggested underdosing was prevalent and 40.5% of children failed to reach achieve therapeutic targets. In the absence of guidelines for voriconazole dosing in children, maintenance doses varied widely in the study. Children less than 2 years old were most likely to have subtherapeutic voriconazole concentrations. Moreover, after TDM-guided dose adjustment the overall probability of target achievement was in fact decreased, from 40.0% to 33.3% (*p* < 0.001; Figure 2b), suggesting inaccurate dose adjustment likely resulting from lack of unified guidance.

Karlsson MO et al. [21] and Neely M et al. [10] illustrated the optimal dose of voriconazole was 7 mg/kg twice daily in children 2 to 12 years old, While, Shima H et al. recommended that for patients younger than 2 years at least 8.5 mg/kg twice daily of voriconazole dose was needed.

However, the targeted media maintenance dose in children under 2, 2–6, 6–12 and 12–18 years old was 5.71, 6.67, 5.08 and 3.31 mg/kg in our study. The dosage was roughly lower than the researches mentioned above. According to the American “VFEND” drug label [3], there is no voriconazole dosing recommendation for children less than 2 years old. For children 2 to 14 years of age weighing less than 50 kg, the maintenance dose of voriconazole was at least 8 mg/kg. The dosage in our study may be lower than the recommended dosage of 8 mg/kg among the children of 2–12 years old. Therefore, further study is needed in the future to evaluate the optimal dosage in children.

In the present study, the primary administration route of voriconazole was oral administration (88.3%) rather than the intravenous route. Variability in oral bioavailability caused by meals and hepatic first-pass effect may account for lower drug exposure [26,27], potentially accounting for sub-therapeutic trough concentrations; 51.1% of the patients received voriconazole prophylactically, however, it is unclear what impact prophylactic administration had on voriconazole concentration in the different age groups and this still needs to be further explored.

Another critical factor affecting voriconazole therapeutic trough concentrations is the polymorphisms of CYP2C19 conferring PM or URM phenotypes. Normally, individuals who are CYP2C19 ultrarapid metabolizers have decreased trough voriconazole concentrations, delaying achievement of target blood concentrations, whereas poor metabolizers have increased trough concentrations and are at increased risk of adverse drug events [17]. The CYP2C19 genotypic and phenotypic variability have been extensively characterized among different ethnic groups but to date has played little role in routine voriconazole dosing. Generally, Caucasians or Africans have a lower proportion of PM metabolizers than Asians (2.00% to 5.00%, 6.00%, and 13.0% to 23.0%, respectively). Furthermore, compared to Caucasians and Africans, Asians are about 4 times more likely to carry the URM CYP2C19*17 allele [28,29,30,31,32]. No CYP2C19*17 alleles were found in our study. It was likely that the sample was too small and too many confounding factors existed. Hicks JK et al. [33] emphasized that a starting dose above 14 mg/kg/day could be recommended for all children except for CYP2C19 extensive metabolizers. Voriconazole dosing regimens in children could be better designed to incorporate age and CYP2C19 genotypes with the aim of dramatically reducing variation of drug trough concentrations. Therefore, available CYP2C19 genotype before the initial administration of voriconazole could improve the accuracy and safety of initial dosing [11].

As for the result of drug interactions, Dolton and Tian, X. et al. [34,35] found that voriconazole concentration increase with the coadministration of PPIs. Dolton et al. [34] also found that coadministration of glucocorticoids may significantly reduce voriconazole concentrations. The effect of PPIs on voriconazole was consistent with our results, while the effect of glucocorticoids was not. However, Blanco-Dorado and Hashemizadeh, Z. et al. [36,37] found no association between voriconazole trough concentration and other factors such as concomitant administration of enzyme inducer, enzyme inhibitor, glucocorticoids, or PPIs. Besides, oral voriconazole has a significant drug interaction with oral tacrolimus with a wide inter-individual variability. Therefore, the interaction between tacrolimus and voriconazole has also been widely concerned [5,6]. The result of our previous study in kidney transplantation recipients was that the combination use of tacrolimus was associated with voriconazole concentration; however, in further multiple linear regression analysis, tacrolimus was not identified as the final influencing factor of voriconazole concentration [38], and the results were similar to those in this study. Taken together, the drugs mentioned above may have potential drug interactions with voriconazole, but the conclusions are inconsistent; it is also necessary to pay attention to the impact of drug combination during voriconazole therapy. Further and high-quality studies are needed to validate the effects of drug interactions on voriconazole concentration and dose.

In our study, final multivariate analysis found that weight, dose before sampling, direct bilirubin, urea nitrogen and phenotypes of CYP2C19 were influencing factors of voriconazole trough concentration, while the results of other studies are compared and shown in Table 6. Researchers focused on different study population to explore the influencing factors. Some factors that were not included in our study may probably have a significant impact on voriconazole concentration, such as C-reactive protein (CRP) level, gamma-glutamyl transferase and IL-6. Besides, we also found some factors that were not included in the previous study, such as direct bilirubin and urea nitrogen; however, different factors may be affected by different study cohorts.

Finally, our study is a single-center retrospective study and limitations in study design and analysis must be considered. Firstly, a small sample size restricts the ability to generate statistically significant results. Secondly, it is a non-intervention study based on real-world data to try to explore voriconazole use in pediatric patients as much as possible, but due to the specificity population of this study population we just collected clinical data without intervention. Thirdly, it is a retrospective study, although we attempted to reduce potential confounding factors, there are still some unavoidable confounding factors such as the differences of individual indicators on baseline. Meanwhile, the lack of uniform guidelines and standardized protocol for voriconazole dose adjustment based on TDM data may have resulted in inconsistent dose adjustment by clinicians. The limited number of CYP2C19 genotypes may have been unable to detect a relationship between trough concentrations and genotypes. In addition, obesity, alkaline phosphatase, co-administered drugs and other single nucleotide polymorphisms such as the SLCO1B3, SLCO1B1, SLC22A6, ABCB1, ABCG2, SLCO3A1, ABCC2, SLC22A1, ABCB11 and NR1I2 genes, have been reported to be associated with decreased metabolism of voriconazole to its inactive N-oxide metabolite [31,32,42,43]; however, these additional confounders were not assessed in our study. Therefore, prospective and multi-center studies are needed to further explore the individualized use of voriconazole in pediatric patients. Specific dose optimization dose is also necessary in future studies.

## 4. Materials and Methods

### 4.1. Patients

The study was conducted retrospectively at the Second Xiangya Hospital of Central South University, examining admissions between 1 January 2016 to 31 December 2020. The study was approved by the ethics committee of the Second Xiangya Hospital of Central South University. In addition, it was registered on ChiCTR.org with the registration number of ChiCTR1900025821(9 September 2019) and conducted according to the Declaration of Helsinki. All patients or their legal guardian provided informed consent for the usage of their clinical data and/or samples.

The inclusion criteria were: children treated with voriconazole and who had trough concentration monitoring performed between age of 2 to 18 years with complete medical records. The exclusion criteria were: a bodyweight of ≥50 kg when patients aged 12 to 18 years (the dosage regimen for this population is consistent with that of the adults according to the instruction book) [20].

### 4.2. Data Collection

From the electronic medical records, the patient demographic and clinical data were collected, including ethnicity, age, sex, body weight (BW), CYP2C19 genotype, underlying disease, treatment indication, site of infection, voriconazole dosing, route of administration, voriconazole trough concentrations, treatment duration, concomitant medications, adverse drug reactions, efficacy, liver function, and kidney function. The concomitant use of medications that were likely to influence voriconazole trough concentrations, such as proton pump inhibitors (PPIs), glucocorticoids and immunosuppressors were also recorded.

### 4.3. Voriconazole Administration and Plasma Trough Concentration Measurement

Voriconazole doses were selected by clinicians according to their experience and/or the drug package insert. The trough concentration was collected three days later if the loading dose was used or five days later after the maintenance dose. Time to steady-state was chosen according to the voriconazole guideline by Chinese Pharmacological Society and the result of two population PK studies [11,18,44,45]. Nurses collected the blood sample within half an hour before the next dose under steady-state conditions.

Voriconazole plasma concentration was measured by an automatic two-dimensional liquid chromatography (2D-HPLC, Demeter Instrument Co., Ltd., Changsha, Hunan, China). The two-dimensional separation conditions consisted of the following: the first-dimensional chromatographic column was FRO C18 (100 mm × 3.0 mm, 5 μm, ANAX); and the flow rate: 1.0 mL/min. The second-dimensional chromatographic column was ASTON HD C18 (150 mm × 4.6 mm, 5 μm, ANAX). The linearity range was 0.35 to 11.26 µg/mL with the quantitative limit of 0.046 µg/mL. The quantitative limit was 0.05 µg/mL. The absolute and relative recovery ranged from 88.2 to 93.6% and 94.2 to 105.3%, respectively. The intra-day and inter-day precisions were 1.94 to 2.22% and 2.15 to 6.78%. The stability of blood sample at room temperature for 8 h and at −20 °C of three repeated freeze-thaw cycles was within ±8.00% and ±10.0%, respectively [46]. Besides, our laboratory performed annual external quality assessment (EQA) to ensure the accuracy of the measurement results. The ideal target trough concentration of voriconazole for both prophylaxis and treatment was set as a range of 1.0 to 5.5 µg/mL [20].

### 4.4. CPY2C19 Genotype and Phenotype Assignment

DNA was separated from the suspending white cells and was purified with the E.Z.N.A^®^ SQ Blood DNA Kit II method. CYP2C19 genotypes were implemented by Sanger dideoxy DNA sequencing method with ABI3730 xl-full automatic sequencing instrument (ABI Co., Carlsbad, CA, USA) Boshang Biotechnology Co. Ltd. in Shanghai, China. According to the definition of the Clinical Pharmacogenetics Implementation Consortium (CPIC) [9], CYP2C19 phenotypes were categorized as several types based on CYP2C19 *1, *2, *3, or *17 allele nomenclature. CYP2C19 phenotypes were classified into five categories: ultrarapid metabolizer (UM, CYP2C19*17/*17), rapid metabolizer (RM, CYP2C19*1/*17), extensive metabolizer (EM, CYP2C19*1/*1), intermediate metabolizer (IM, CYP2C19*1/*2, CYP2C19*1/*3, CYP2C19*2/*17) and poor metabolizer (PM, CYP2C19*2/*2, CYP2C19*2/*3, CYP2C19*3/*3).

### 4.5. Outcome and Safety Assessment

The definition of IFI was in accordance with the European Organization for Research and Treatment of Cancer/Invasive Fungal Infections Cooperative Group and the National Institute of Allergy and Infectious Diseases Mycoses Study Group (EORTC/MSG) [24]. Additionally, during the study, voriconazole-attributable adverse reactions were recorded based on the Common Terminology Criteria for Adverse Events (CTCAE) [26].

### 4.6. Statistical Analysis

Continuous variables were expressed as median (range: quartiles). Categorical data were reported as frequencies and percentages. For quantitative data, the normality was tested using the Shapiro–Wilk test. Student’s *t*-test or Mann–Whitney test was selected according to the result of normality. Chi-square test or Fisher exact tests were selected to test the enumeration data. A two-tailed test with a *p*-value ≤ 0.05 was considered statistically significant. Results were given as point estimates or 95% confidence intervals. Subsequently, the determinations of voriconazole trough concentration were then analyzed using multiple linear regression. The phenotype of CYP2C19 was set as a dummy variable. We conducted the spearman correlations and point-biserial correlation analyses to select factors correlated first. A variance inflation factor (VIF) of >5 was considered indicative of multicollinearity. We conducted all analyses using IBM SPSS Statistics version 25 (IBM, New York, NY, USA) and drew the figures using GraphPad Prism version 8 (San Diego, CA, USA).

## 5. Conclusions

In conclusion, younger children less than 12 years old tend to have a higher inter- and intra-individual variability than the children over 12 years old and should be a focus when prescribing voriconazole. Meanwhile, children less than 2 years old likely need to have a higher dosage regime. The maintenance doses required to achieve the target concentration for children less than 2, 2 to 6, 6 to 12, and 12 to 18 years old years old were approximately about 5.71, 6.67, 5.08 and 3.31 $\mathrm{mg}\cdot {\mathrm{kg}}^{-1}/12\;h$ respectively. Voriconazole maintenance doses under the target concentration tend to be lower than drug label recommended, but this still requires further study. In Chinese chilren, age, body weight, dose, direct bilirubin, urea nitrogen and phenotypes of CYP2C19 were found to be influencing factors of voriconazole concentration.

These factors can explain 36.2% of the variability in voriconazole trough concentration. The influence of these factors should be taken into consideration during voriconazole use.

## Figures and Tables

**Figure 1 antibiotics-10-01542-f001:**
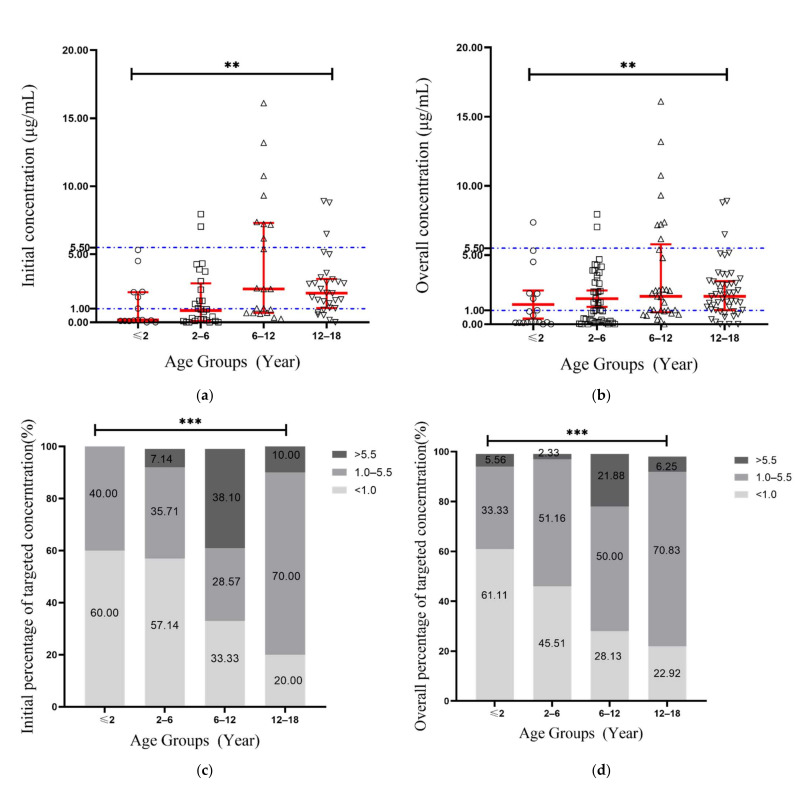
Concentration distribution and percentage of target concentration. (**a**) Scatter dot plot of the initial trough concentration among four age groups; error bars indicate the median and interquartile range; (**b**) Scatter dot plot of the overall trough concentration among four age groups; error bars indicate the median and interquartile range; (**c**) Initial probability of achieving targeted concentration in four age groups; (**d**) Overall probability of achieving targeted concentration in four age groups.** *p* < 0.01; *** *p* < 0.001.

**Figure 2 antibiotics-10-01542-f002:**
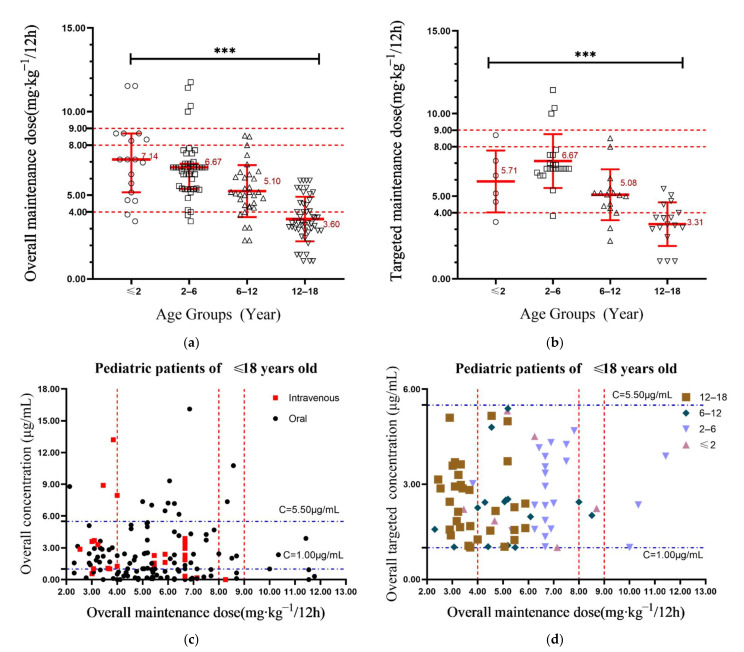
(**a**) The overall maintenance dosage of four age groups; error bars indicates the median and interquartile range); (**b**) The overall targeted maintenance dosage of four age groups; error bars indicates the median and interquartile range); (**c**) Scatter diagram of the overall voriconazole trough concentration at different maintenance dosage; (**d**) Scatter diagram of the overall voriconazole trough concentration at different maintenance dosage under targeted concentration of 1.0–5.5 µg/mL. The maintenance dosage is weight-adjusted. The red dotted line indicates the recommended dosage by the drug label; *** *p* < 0.001.

**Figure 3 antibiotics-10-01542-f003:**
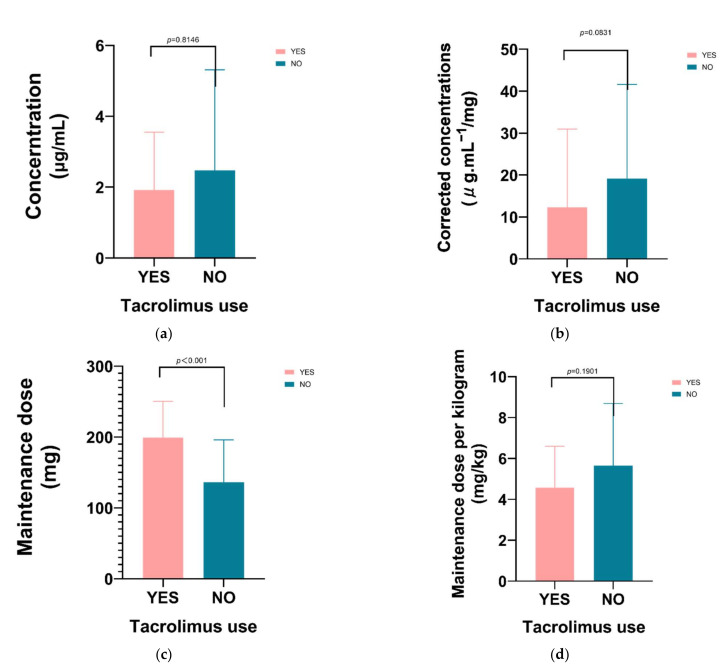
The influence of tacrolimus use was only significant on voriconazole maintenance dose (*p* < 0.001). (**a**) The influence of tacrolimus use on voriconazole concentration; (**b**) The influence of tacrolimus use on corrected voriconazole concentration (concentration/dose); (**c**) The influence of tacrolimus use on maintenance dose; (**d**) The influence of tacrolimus use on unit kilogram maintenance dose.

**Figure 4 antibiotics-10-01542-f004:**
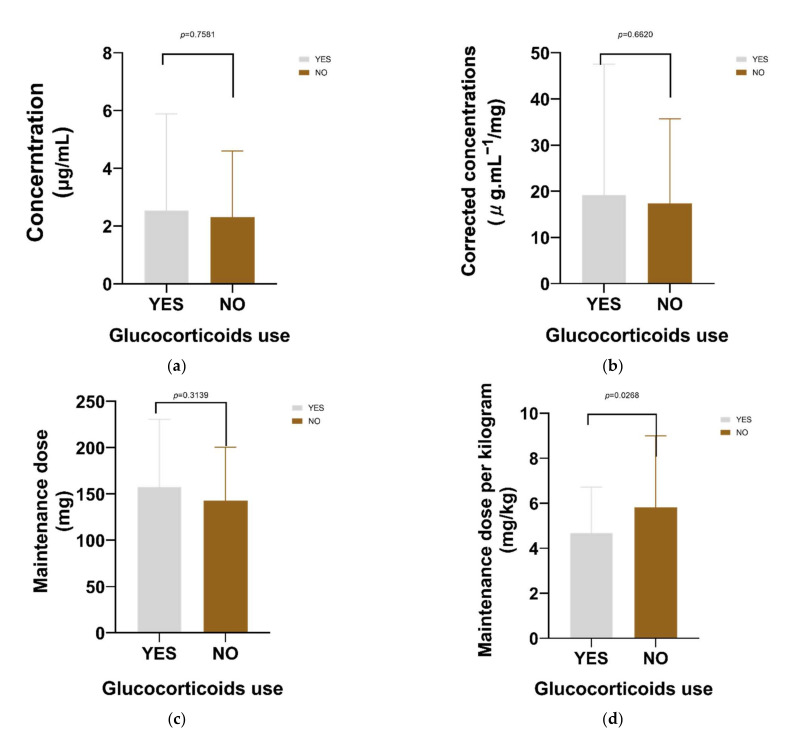
The influence of glucocorticoids use was only significant on unit kilogram maintenance dose of voriconazole (*p* = 0.0268). (**a**) The influence of glucocorticoids use on voriconazole concentration; (**b**) The influence of glucocorticoids use on corrected voriconazole concentration (concentration/dose); (**c**) The influence of glucocorticoids use on maintenance dose; (**d**) The influence of glucocorticoids use on unit kilogram maintenance dose.

**Figure 5 antibiotics-10-01542-f005:**
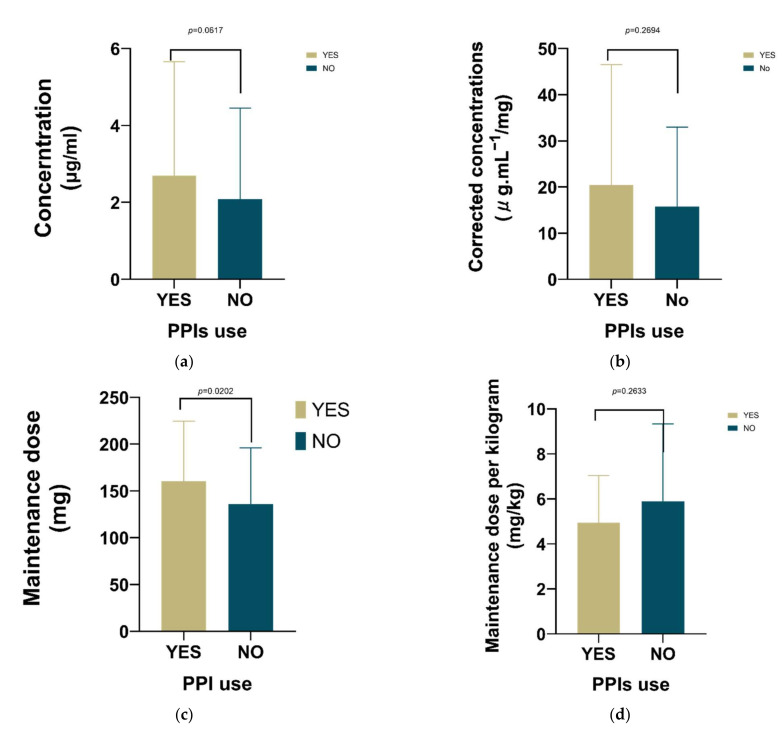
The influence of PPIs was only significant on the maintenance dose of voriconazole (*p* = 0.0202). (**a**) The influence of PPI’s use on voriconazole concentration; (**b**) The influence of PPI’s use on corrected voriconazole concentration (concentration/dose); (**c**) The influence of PPIs use on maintenance dose; (**d**) The influence of PPI’s use on unit kilogram maintenance dose.

**Figure 6 antibiotics-10-01542-f006:**
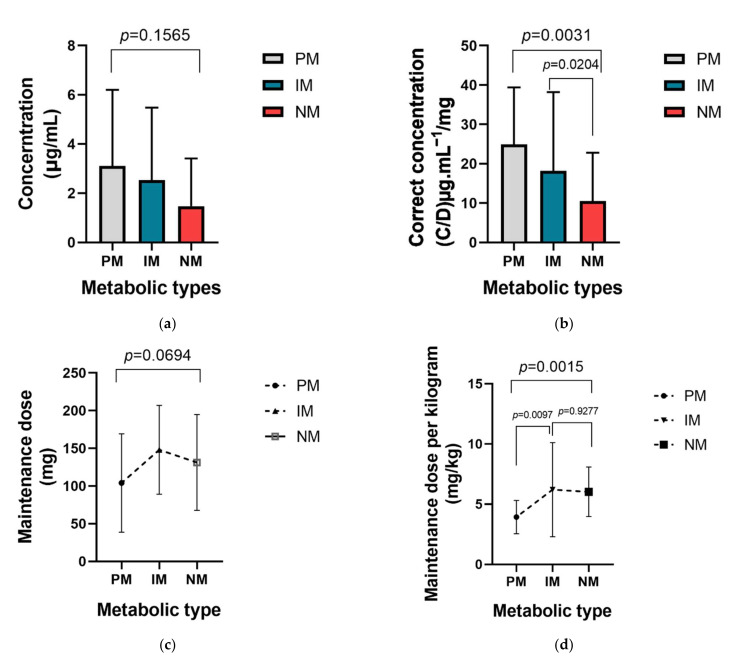
The influence of CYP2C19 metabolic types on were statistically significant not only on the corrected voriconazole concentration but also on the unit kilogram maintenance dose. (**a**) The influence of CYP2C19 metabolic types on voriconazole concentration; (**b**) The influence of CYP2C19 metabolic types on corrected voriconazole concentration (concentration/dose); (**c**) The influence of CYP2C19 metabolic types on maintenance dose; (**d**) The influence of CYP2C19 metabolic types on unit kilogram maintenance dose.

**Figure 7 antibiotics-10-01542-f007:**
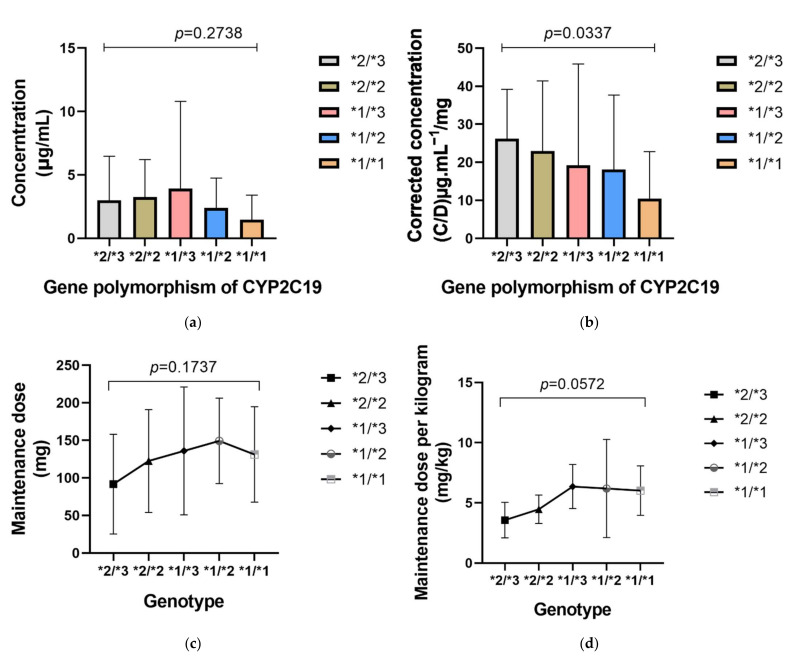
The influence of CYP2C19 gene polymorphism was statistically significant on the corrected voriconazole concentration, while its influence on the unit kilogram maintenance dose was nearly significant. (**a**) The influence of CYP2C19 gene polymorphism on voriconazole concentration; (**b**) The influence of CYP2C19 gene polymorphism on corrected voriconazole concentration (concentration/dose); (**c**) The influence of CYP2C19 gene polymorphism on maintenance dose; (**d**) The influence of CYP2C19 gene polymorphism on unit kilogram maintenance dose.

**Table 1 antibiotics-10-01542-t001:** Patients’ demographic data and clinical characteristics.

Characteristics	≤2 y	2 y–6 y	6 y–12 y	12 y–18 y	*p*
Sex(male)	8 (53.3%)	12 (42.9%)	13 (61.9%)	16 (53.3%)	0.616
Weight(kg)	11.5 (9.20–14.0)	16.0 (14.5–18.0)	30 (23.5–35.8)	54.7 (43.6–61.8)	<0.0001 *
**IFI diagnosis**	15 (16.0%)	28 (29.8%)	21 (22.3%)	30 (31.9%)	
Proven	2 (13.3%)	5 (17.9%)	2 (9.50%)	6 (20.0%)	0.3535
Probable	8 (53.3%)	8 (28.6%)	9 (42.9%)	6 (20.0%)	
Possible	5 (33.3%)	15 (53.6%)	10 (47.6%)	18 (60.0%)	
**Infection site**					
Pulmonary infection	13 (86.7%)	11 (39.3%)	9 (42.9%)	21 (70.0%)	0.1600
Bloodstream infection	1 (6.67%)	7 (25.0%)	5 (23.8%)	3 (10.0%)	
Perineum infection	0 (0.00%)	1 (3.57%)	0 (0.00%)	0 (0.00%)	
Oral infection	0 (0.00%)	1 (3.57%)	0 (0.00%)	0 (0.00%)	
Unknown	1 (6.67%)	8 (28.6%)	7 (33.3%)	6 (20.0%)	
**Treatment indication**					
Therapeutic	3 (20.0%)	6 (21.4%)	2 (9.5%)	8 (26.7%)	0.6782
Empirical	6 (40.0%)	8 (28.6%)	7 (33.3%)	6 (20.0%)	
Prophylactic	6 (40.0%)	14 (50.0%)	12 (57.1%)	16 (53.3%)	
**Drug combination**					
PPIs ^2^	3 (20.0%)	12 (42.9%)	8 (38.1%)	18 (60.0%)	0.0737
Glucocorticoids	5 (33.3%)	5 (17.9%)	6 (28.6%)	11 (36.7%)	0.3692
Immunosuppressive drugs	0 (0.00%)	3 (10.7%)	0 (0.00%)	6 (80.0%)	0.0546
**CYP2C19 ^1^ Genotype**					
1/*1	1 (6.67%)	12 (42.9%)	4 (19.1%)	2 (6.7%)	0.0004 *
1/*2	6 (40.0%)	5 (17.9%)	7 (33.3%)	10 (33.3%)	
1/*3	3 (20.0%)	0 (0.00%)	1 (4.76%)	0 (0.00%)	
2/*2	1 (6.67%)	0 (0.00%)	1 (4.76%)	1 (3.33%)	
2/*3	0 (0.00%)	0 (0.00%)	3 (14.29%)	0 (0.00%)	
Unknown	4 (26.7%)	11 (39.3%)	5 (23.8%)	17 (56.7%)	
Alanine transaminase	35.86 ± 24.21	35.45 (12.15–60.35)	33.30 (19.15–49.90)	16.70 (8.73–32.08)	0.0829
Aspartate aminotransferase	31.80 (27.80–49.40)	31.25 (17.78–43.93)	27.70 (17.45–40.00)	15.00 (11.48–33.90)	0.0033 *
Total bilirubin	8.20 (5.70–10.00)	9.75 (6.25–19.18)	13.77 ± 7.77	10.85 (7.95–15.50)	0.1392
Direct bilirubin	3.00 (2.10–3.90)	3.85 (2.55–7.63)	4.80 (3.10–7.20)	4.40 (3.13–6.65)	0.0718
Albumin	35.29 ± 4.78	33.22 ± 5.45	33.41 ± 3.72	33.40 (29.65–37.25)	0.5002
Blood Urea Nitrogen	2.52 (1.72–3.79)	4.02 (2.92–5.86)	4.67 (2.56–7.11)	5.59 (4.3–7.50)	0.0032 *
Creatinine	18.44 ± 4.19	20.00 (18.15–27.20)	33.40 (25.45–36.95)	58.05 (43.43–95.38)	<0.0001 *
Uric Acid	237.1 ± 100.5	228.0 ± 104.5	206.0 (162.8–295.4)	229.4 (161.5–311.0)	0.9342

Normality is tested by Shapiro–Wilk test. The normality results are expressed as mean ± standard deviation, while the median (quartile) was used for non-normal results. Enumeration data is expressed in percentage form. ^1^ CYP2C19, cytochrome P450 2C19; ^2^ PPIs: proton pump inhibitors; * The distinction was statistically significant, at the level of 0.05 (double tail).

**Table 2 antibiotics-10-01542-t002:** Patients’ voriconazole administration and TDM data.

Characteristics	≤2 y	2 y–6 y	6 y–12 y	12 y–18 y	*p*
Administration route (Oral)	12 (85.7%)	27 (96.4%)	20 (95.2%)	24 (80.0%)	0.1615
Initial Ctrough (µg/mL)	0.17 (0.11–2.1)	0.87 (0.11–2.86)	2.45 (0.69–7.30)	2.14 (1.05–3.19)	0.0014 *
<1.0	9 (60.0%)	16 (57.1%)	7 (33.3%)	6 (20.0%)	0.004 *
1.0−5.5	6 (40.0%)	10 (35.7%)	6 (28.6%)	21 (70.0%)	
>5.5	0 (0.00%)	2 (7.14%)	8 (38.1%)	3 (10.0%)	
VRC Ctrough (µg/mL)	0.18 (0.11–2.21)	1.19 (0.22–3.27)	2.02 (0.86–5.78)	2.02 (1.02–3.12)	0.0096
Dose adjustment	4 (26.7%)	7 (25.0%)	8 (38.1%)	2 (6.67%)	0.0568
Initial dose ($\mathrm{mg}\cdot {\mathrm{kg}}^{-1}/12\;h$)	7.10 (4.70–8.70)	6.30 (5.40–6.90)	5.20 (4.35–6.40)	3.35 (3.08–4.13)	<0.0001 *
Overall dose ($\mathrm{mg}\cdot {\mathrm{kg}}^{-1}/12\;h$)	7.14 (5.17–8.70)	6.67 (5.38–6.90)	5.10 (4.29–6.07)	3.60 (2.92–4.62)	<0.0001 *
Target initial dose ($\mathrm{mg}\cdot {\mathrm{kg}}^{-1}/12\;h$)	5.75 (4.38–7.50)	6.90 (6.60–7.50)	5.10 (4.85–5.90)	3.30 (3.15–4.5)	<0.0001 *
Target overall dose ($\mathrm{mg}\cdot {\mathrm{kg}}^{-1}/12\;h$)	5.71 (4.36–7.53)	6.67 (6.61–7.50)	5.09 (4.32–5.41)	3.31 (2.77–4.25)	<0.0001 *

Ctrough: the voriconazole trough concentration; * The distinction was statistically significant at the level of 0.05 (double tail). Bonferroni adjustment was used to make a pairwise comparison analysis.

**Table 3 antibiotics-10-01542-t003:** Bonferroni adjustment result of pairwise comparison.

Pairwise Comparison	Adjusted *p* Value			
	Initial Concentration (µg/mL)	Overall Concentration (µg/mL)	Overall Maintenance Dose(mg/kg)	Targeted Maintenance Dose (mg/kg)
≤2 y vs. 2 y–6 y	1.00	1.00	1.00	1.00
≤2 y vs. 6 y–12 y	0.012 *	0.023 *	0.023 *	1.00
≤2 y vs. 12 y–18 y	0.035 *	0.042 *	<0.001 *	0.047 *
2 y–6 y vs. 6 y–12 y	0.028 *	0.258	0.024 *	0.011 *
2 y–6 y vs. 12 y–18 y	0.087	0.489	<0.001 *	<0.001 *
6 y–12 y vs. 12 y–18 y	1.00	1.00	0.001 *	0.094

* The distinction was statistically significant at the level of 0.05 (double tail); *p* values have been adjusted for multiple test results.

**Table 4 antibiotics-10-01542-t004:** Correlation analysis of voriconazole trough concentration.

Demographic Variable	Coefficient Index	*p*-Value
Age	0.263 *	0.005
Weight (kg)	0.288 *	0.001
BMI	0.179	0.056
Height	0.277 *	0.003
First maintenance dose	−0.062	0.770
Dose before sampling	0.266 *	<0.001
Physiological and biochemical indexes		
Aspartate aminotransferase	−0.207 *	0.017
Alanine transaminase	0.071	0.421
Total Bilirubin	0.207 *	0.017
Direct Bilirubin	0.175 *	0.045
Albumin	−0.199 *	0.022
Urea nitrogen	0.216 *	0.013
Creatinine	0.267 *	0.002
Administration route	0.046	0.585
Proton pump inhibitors use	0.136	0.128
Immunosuppressants use	−0.020	0.813
Glucocorticoid use	−0.026	0.758
CYP2C19 phenotypes	−0.263 *	0.011
CYP2C19 genotype	−0.236 *	0.024

* The variables is significant, at the level of 0.05 (double tail).

**Table 5 antibiotics-10-01542-t005:** Multiple linear regression analysis of voriconazole trough concentration determinants.

	Coefficient	T	*p* Value	VIF
Weight	−0.050	−2.398	0.019 *	2.033
Dose before sampling	0.033	5.407	0.000 *	2.070
Direct Bilirubin	0.055	3.976	0.000 *	1.114
Urea nitrogen	0.216	2.109	0.038 *	1.135
CYP2C19 phenotype (IM)	−1.789	−2.042	0.045 *	2.863
CYP2C19 phenotype (NM)	−2.020	−2.194	0.031 *	2.794
Constant value	−0.655	−0.600	0.550	
F		8.551		
*p*		<0.001 *		
R^2^		0.362		
Dependent variable: voriconazole trough concentration				

* The variables were significant, at the level of 0.05 (double tail); CYP2C19, cytochrome P450 2C19. The dummy variable is set during the analysis and the result is compared with the CYP2C19 phenotype (PM) group. The maintenance dosage is weight-adjusted. The *p* value for each group is indicated above the figure.

**Table 6 antibiotics-10-01542-t006:** Comparison of multiple linear regression models and voriconazole concentration influencing factors.

Final Factors	References	No. of Patients	No. of Concentration	Study Cohort	R^2^
Weight, voriconazole dose, direct bilirubin, urea nitrogen and CYP2C19 phenotypes	result of this study	94	145	Pediatric patients	0.362
Age, CYP2C19 phenotype, PPIs	Tian et al., 2021 [35]	108	348	Pediatric patients	0.234
CYP2C19 phenotypes, hemoglobin, platelet count, PPIs	Zhao et al., 2021 [38]	93	213	Kidney transplantation recipients	0.336
C-reactive protein (CRP) level, albumin, glucocorticoid	Dote et al., 2019 [39]	63	77	Elderly patients	/
Sex, daily dose, CYP2C19 genotyping, platelet, and MELD score	Zhao et al., 2019 [40]	43	144	Child–Pugh class C patients	0.348
Age, gamma-glutamyl transferase, IL-6, PPIs, CYP2C19 phenotypes	Mafuru et al., 2019 [41]	113	250	Patients With Hematologic Disorders	0.290
Weight, oral voriconazole, phenytoin or rifampin, and PPIs	Dolton et al., 2012 [34]	201	783	Adults	/
CYP2C19 phenotype	Blanco et al., 2019 [36]	78	/	Adults	/
Administration routes, PPIs	Lin Hu et al., 2018 [24]	42	138	Pediatric patients	0.553

## Data Availability

I declare that my research data is available. The data will include (but is not limited to): raw data, processed data, software, algorithms. The researchers can contact me if they need the research data.

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
