# Peer review of "Factors Affecting Voriconazole Trough Concentration and Optimal Maintenance Voriconazole Dose in Chinese Children"

_antibiotics, 2021, doi:10.3390/antibiotics10121542_

Round 1

Reviewer 1 Report

No further comments, thank you for your detailed revisions and explanations.

Author Response

 Thanks very much for the recognition of our work, and your affirmation encourages us a lot.

Reviewer 2 Report

An interesting retrospective study of TDM of voriconazole in children is still quite unknown. Due to the fact that recommendations are only provided for caucasian children, further investigation in non-caucasian children of genetic and other factors that may influence the concentration of voriconazole is necessary. As is so often the case with retrospective studies, there are factors that are difficult to control that could potentially affect the overall results. Notwithstanding that, the manuscript was interesting to read, although there are a lot of dates presented and therefore it is not always easy to follow

Line85-86 Probably missing a point in the sentence

The indications for which the children were treated with voriconazole were probably not invasive fungal infections (mouth, perineum, bloodstream) and rather mean Candida colonization/infection. The authors would therefore better explain what oral and especially perineal infections represent.

51% of patients received voriconazole prophylactically. It is not clear what impact prophylactic administration had on actual TDM in the different age groups. I assume that these patients had been administered voriconazole for some time

Section 4.3 states that "trough concentration was collected three days later if the loading dose was used or five days later after the maintenance dose. However, in Table2, the collection times are very variable

For a better overview and readability of the tables, they should be modified

Avoid reusing (P) data several times in the text and figures. example1 some of the data in Table 2 are reused again in Figure 1. example 2 table6 Use either one.
Correct error in title of figure 1

It is known that the variability of the trough level of voriconazole can be large, however, in this study, there were only 7 patients in which the TDM could be performed multiple times. It can therefore be assumed that the variability is also high in children. The fact that the coefficient of variation is so large with only 2 drops is not surprising. This paragraph could be shortened

How confident are the authors that with voriconazole administered orally, that the correct guidelines for administration and compliance were followed

Line 327-329 The result of our previous study in kidney transplantation recipients was that tacrolimus was correlated with voriconazole trough concentration, but it did not enter into the final multiple linear regression model: "tacrolimus was correlated with voriconazole" This sentence is not clear to me

Author Response

Comments and Suggestions for Authors

An interesting retrospective study of TDM of voriconazole in children is still quite unknown. Due to the fact that recommendations are only provided for caucasian children, further investigation in non-caucasian children of genetic and other factors that may influence the concentration of voriconazole is necessary. As is so often the case with retrospective studies, there are factors that are difficult to control that could potentially affect the overall results. Notwithstanding that, the manuscript was interesting to read, although there are a lot of dates presented and therefore it is not always easy to follow.

The authors’ response: Thanks very much for the positive comments about this manuscript, and it is your affirmation that encourages us a lot. Referring to your suggestions, we have displayed point-by-point responses in following context.

Point 1: Line85-86 Probably missing a point in the sentence.

The authors’ response: Thanks for your careful check. We have added a point in the sentence. Meanwhile, after reading through the context, some sentences have also been further modified.

(Clean version: Line 88-91; Trace version: Line 88-91).

Point 2: The indications for which the children were treated with voriconazole were probably not invasive fungal infections (mouth, perineum, bloodstream) and rather mean Candida colonization/infection. The authors would therefore better explain what oral and especially perineal infections represent.

The authors’ response: Thanks for the professional comments. I have added the indications for voriconazole in the section of introduction. But I may not have fully understood the reviewer's opinion. If you have any other questions about our reply, please feel free to contact me.

The revised manuscript is as follows: Voriconazole could be used for the prophylaxis and treatment of invasive aspergillosis and candidiasis in non-neutropenia patients. Some serious invasive infections caused by fluconazole-resistant candida (including Candida kronella) and the genera of Actinomy-cetes and Fusarium can also be treated by voriconazole.

(Clean version: Line 40-44; Trace version: Line 40-44).

Point 3: 51% of patients received voriconazole prophylactically. It is not clear what impact prophylactic administration had on actual TDM in the different age groups. I assume that these patients had been administered voriconazole for some time

The authors’ response: Thanks for the professional comments. This is a good question that needs to be further explored, and we will conduct further analysis in the following study. We have also added relevant descriptions in the discussion section. Meanwhile, the administration of voriconazole was all recorded from the first dose instead of using voriconazole for some time already.

The added content was: 51.1% of the patients received voriconazole prophylactically. However, it is still unclear what impact prophylactic administration had on voriconazole concentration in the different age groups. It needs to be further explored.

(Clean version: Line 301-304; Trace version: Line 307-310).

Point 4: Section 4.3 states that "trough concentration was collected three days later if the loading dose was used or five days later after the maintenance dose. However, in Table2, the collection times are very variable. For a better overview and readability of the tables, they should be modified

The authors’ response: Thanks for your suggestion. Trough concentrations were collected as described above. It was exactly collected three days later if the loading dose was used or five days later after the maintenance dose. It was not just the third or fifth day that concentration could be collected. That’s why the collection times varied from 3 to 19 days. Meanwhile, to make the manuscript more readable, we have modified Table 2.

(Clean version: Line 136-137; Trace version: Line 136-137).

Point 5: Avoid reusing (P) data several times in the text and figures. example1 some of the data in Table 2 are reused again in Figure 1. example 2 table6 Use either one.

Correct error in title of figure 1

The authors’ response: Thanks for your suggestion. The repetitive P values have been removed from the figure. Both Figure 1 and Figure 2 have been revised. Other repeated p-values have also been removed. Deleted duplicate P values have been highlighted in the revised version. (Trace version: Line 139-143/144-152/157-163/192-194/208-210/219-221/232-234/248/253/266/349-351/357).

The title of figure 1 has been also revised. (Clean version: Line 139; Trace version: Line 139).

Point 6: It is known that the variability of the trough level of voriconazole can be large, however, in this study, there were only 7 patients in which the TDM could be performed multiple times. It can therefore be assumed that the variability is also high in children. The fact that the coefficient of variation is so large with only 2 drops is not surprising. This paragraph could be shortened.

The authors’ response: Thanks for your suggestion. According to your comments, we have shortened the description about the great variability of voriconazole administration in children. The revised content is as follows: The coefficient of variation ranged from 17.4% to 143.0%. The inter-individual coefficient of variation ranged from 1.68% to 678.5%. Scatter diagrams and interpatient variability of overall voriconazole trough concentration at different weight-adjusted doses and overall maintenance dose within the target concentration range were shown in Figure.2c and Figure.2d, respectively.  (Clean version: Line 165-169; Trace version: Line 168-178).

Point 7: How confident are the authors that with voriconazole administered orally, that the correct guidelines for administration and compliance were followed.

The authors’ response: Thanks for the valuable comments. Although this was a retrospective study, our hospital had a good management model for patients' drug administration and compliance. Doctors in our hospital have prepared a medication schedule for patients according the instructions and guidelines. Almost all the patients took the medicine strictly according to the prescribed time and dosage shown on the medication schedule. Therefore, we're confident about that.

Point 8: Line 327-329 The result of our previous study in kidney transplantation recipients was that tacrolimus was correlated with voriconazole trough concentration, but it did not enter into the final multiple linear regression model: "tacrolimus was correlated with voriconazole" This sentence is not clear to me.

The authors’ response: Thanks for the valuable comments. Univariate correlation analysis is needed before multivariate regression. The sentence means that the combination use of tacrolimus was associated with voriconazole concentration during the first step analysis. However, in further multiple linear regression analysis, tacrolimus was not the final influencing factor of voriconazole concentration. To avoid confusion, we have revised the content in the manuscript. The revised content is as follows: The result of our previous study in kidney transplantation recipients was that the combination use of tacrolimus was associated with voriconazole concentration. However, in further multiple linear regression analysis, tacrolimus was not identified as the final influencing factor of voriconazole concentration. (Clean version: Line 332-336; Trace version: Line 337-343).

Reviewer 3 Report

This paper presents a retrospective study on the maintenance dose of voriconazole in children, in function of a number of physiological parameter and of genotype of CYP2C9, the principal P450 metabolizing that drug.

The study is well presented and easy to follow.

However when looking at the graphic it is evident that the error bars are quite big, and that it will not be easy to statistically demonstrate the influence of the major parameters.

The statiscical part seems to be well done (however I am not a statistician) but the size of the population 94 cases with a big dispersion of age, body weight, CYP2C19 genotypes …) make think difficult.

Thus my impression is that the authors have done their best to try to show some significant trends.

The paper is well written, and could be published allowing other scientist working on similar populations to merge their data and obtain more significant trends.

The study does not show unexpected results.

Author Response

Author's Reply to the Review Report (Reviewer 3)

This paper presents a retrospective study on the maintenance dose of voriconazole in children, in function of a number of physiological parameter and of genotype of CYP2C9, the principal P450 metabolizing that drug. The study is well presented and easy to follow. However, when looking at the graphic it is evident that the error bars are quite big, and that it will not be easy to statistically demonstrate the influence of the major parameters. The statiscical part seems to be well done (however I am not a statistician) but the size of the population 94 cases with a big dispersion of age, body weight, CYP2C19 genotypes …) make think difficult. Thus, my impression is that the authors have done their best to try to show some significant trends. The paper is well written, and could be published allowing other scientist working on similar populations to merge their data and obtain more significant trends. The study does not show unexpected results.

The authors’ response: Thanks very much for the positive comments about this manuscript. Your affirmation encourages us a lot. We are also working on further prospective studies to further explore voriconazole use in pediatric patients. However, about your doubt on the big error bars, we want to explain that this was a clinical study and it was indeed different from the experimental and scientific studies with high requirements on the precision and accuracy of the results. Meanwhile, during the analysis, we have considered the normal distribution of data and the specific requirements for p-value calculation. That’s why, most of the results were presented as the median and quartiles instead of mean and standard deviation. Therefore, the research results are relatively objective and credible despite of the limited sample size and multiple variables. In addition, the large error bars were probably caused by the large inter-individual and intra-individual variability of the study cohort. However, I’m not sure whether my explanation is able to respond to your doubts. If you still have any other questions about our reply, please feel free to contact me.

This manuscript is a resubmission of an earlier submission. The following is a list of the peer review reports and author responses from that submission.

Round 1

Reviewer 1 Report

Review of “Factors affecting voriconazole trough concentration and optimal maintenance voriconazole dose in Chinese children”.  The authors attempt to address a gap in our knowledge of voriconazole PK – dosing in Chinese pediatric patients. However, following my review I have several major concerns and minor concerns about this study which significantly hamper the quality of the article. One major concern is that there are some many confounding variables and small sample size, so is it possible to draw any meaningful conclusions from the data?

Abstract:

  1. Line 24 – units are inappropriately stated
  2. Lines 30-31 – statement is not supported by the data and thus should be removed
  3. Lines 31-32 – this sentence is a re-statement of what is already known, as there are published guidelines to direct clinical practice
  4. Line 29 – after scrutinizing the authors findings, I don’t believe they found anything significant about CYP2C19 pharmacogenetics (see comments below about data interpretation)

Introduction:

  1. Lines 44-45 – the word narrow is unnecessarily repeated twice
  2. Line 50 – the authors should consider adding the reference for the published Voriconazole-CYP2C19 guideline (Moriyama et al.)
  3. The authors are repetitive in certain types of information that interrupts the flow of the material (e.g., how many times do you need to state the various factors influencing voriconazole PK? how many times do you need to state that PK is poorly known in children?)
  4. Line 59 – do you mean linear or non-linear PK?
  5. Line 62 – what is meant by ‘American’? do you mean FDA-recommended?
  6. Line 65 – ref #19 is not appropriate here; should it be ref#20?

Materials and Methods:

  1. Reference provided on line 308 is not accessible, so please revise or delete from article.
  2. Lines 319-322 – voriconazole has many possible drug interactions, so why only these few medications considered? further confusing is that tacrolimus is not known to influence voriconazole levels, but voriconazole is thought to perturb tacrolimus elimination.  So why was tacrolimus selected?
  3. Line 326 – Time to steady-state is still 4-5 half-lives with continued administration, so why was trough collected 3-days after loading dose and 5-days after maintenance dose?
  4. Section 4.3 – what is the LLOQ? authors should provide a bit more details on the analytical method.
  5. Section 4.3 – please explain the specific phenotype for each of the star alleles.

Results:

  1. Line 96 – round percentage to 3 significant digits
  2. Line 58 – do mean patients less than 2 y/o have greater variability?
  3. There is significant variability in the study population (disease state, dose, day of collection, age group) that makes interpretation of drug levels difficult if not impossible.
  4. Figure 1a – The legend needs to define what the plot and error bars are referring to. Based on the figure it seems impossible that there are significant differences between age groups (they appear to all overlap significantly). What is the n-values for each group?  Are you including multiple troughs for the same patient in each group?
  5. Figure 2 – Units in 2a and 2b are incorrectly stated.
  6. Table 3 – results show that methylprednisolone (glucocorticoid) had no correlation with voriconazole troughs. Yet in the Abstract (lines 30-31) the authors mention that concomitant use of glucocorticoids is of concern.  What is the basis for this statement?
  7. Figures 3-7 – what is meant by the word ‘distinction’? it seems out of place and needs to be replaced
  8. All figures – The legends could be much improved by providing more information on the data presented. I have concerns about the statistical analysis for all of these figures (see previous comment about Figure 1a). Double-check all units in axis labels as there are many errors.
  9. Figure 3, 7a and 7b – Printed in black and white the slight color variations are very hard to interpret.

Discussion:

  1. Line 257 – what is meant by ‘media’?
  2. Line 297 – what is ‘56’ referring to?
  3. There is no discussion on why tacrolimus use was associated with higher voriconazole maintenance doses, if this is even a true finding.
  4. Due to the enormity of confounding factors in this study, it is next to impossible to draw meaningful conclusions from the data.

Reviewer 2 Report

The authors assessed the factors affecting the voriconazole through concentration and optimal maintenance dose for it in Chinese children. This paper is interesting and policy-based. It should be published.  I have some concerns, which are given below:

  • As the study was retrospective, what was the validity of the lab values, tests/procedure.
  • The introduction needs revision, as there is a certain ambiguity in the statements.
  • In the method section, it should be included that how confounding variables were controlled, and for repeating chi-square the Bonferroni adjustment/correction should be made.

Abstract:

Line 20: write as "1th"  apply this format throughout the manuscript

Introduction:

Line 40: rewrite the sentence as its statement is confusing " voriconazole has a narrow therapeutic index is narrow"

Line 53: provide the references of the respective studies.

Line 58: This sentence is repeating. kindly remove it.

Line 59: in the first paragraph of the introduction the authors quoted that voriconazole showed complex non-linear kinetics but in this statement the authors stated about the variable linear kinetics of this drug. this two statement is contradictory.

Result:

Table.1: Patient demographic data & clinical characteristics

Colum .1 x Row .1 (Sex male/Female) : is this variable represent both sex or what deos it mean

Colum .1 x Row .1 (Sex male/Female) : p value equal : what does this p value represents?

Table. 2: Patients’ voriconazole administration TDM data”

if authors are using the chi-square test, then it should be adjusted by using Bonferroni Adjustment, through the results.

Method:

4.6 statistical Analysis:

Line 355: The p-value should be equal to or less than 0.05